# The Identification of a Novel Pathogenic Variant in the GATA6 Gene in a Child with Neonatal Diabetes

**DOI:** 10.3390/ijms252211998

**Published:** 2024-11-08

**Authors:** Elena A. Sechko, Maria P. Koltakova, Rita I. Khusainova, Ildar R. Minniakhmetov, Dmitry N. Laptev

**Affiliations:** Endocrinology Research Centre, Moscow 117292, Russiakhusainova.rita@endocrincentr.ru (R.I.K.); minniakhmetov.ildar@endocrincentr.ru (I.R.M.);

**Keywords:** diabetes mellitus in children, monogenic diabetes, neonatal diabetes mellitus, GATA6, pancreatic hypoplasia

## Abstract

GATA6 syndrome is a rare monogenic disorder caused by heterozygous variants in the gene *GATA6*, which controls the early embryonic differentiation of germ layers and the development of different organs. We present the results of the 7-year follow-up of a child with this syndrome as well as the following conditions: diabetes mellitus, exocrine pancreatic insufficiency, gallbladder atresia, and congenital heart disease (CHD). At birth, the patient was diagnosed with neonatal diabetes mellitus (NDM) associated with heart (mitral valve prolapse) and gastrointestinal abnormalities (gallbladder atresia). Diabetes remitted within weeks and relapsed at the age of 2. We identified a de novo variant of a 4-nucleotide deletion (c.1302+4_1302+7del), previously unreported in the literature, in the donor splicing site of exon 3 of the *GATA6* gene in a heterozygous state. Screening for other possible components of GATA6 syndrome revealed exocrine pancreatic insufficiency, and pancreatic enzyme replacement therapy resulted in improved dyspeptic symptoms, and growth rates increased. In addition, the patient was diagnosed with autoimmune thyroiditis and progressive myopia.

## 1. Introduction

Diabetes mellitus is a clinically and genetically heterogeneous group of diseases. Monogenic diabetes is caused by a single gene defect, diagnosed in 1–2% of pediatric cases [1]. Currently, more than 40 genetic variants of monogenic diabetes are described, among which maturity-onset diabetes of the young (MODY) is the most common, followed by neonatal diabetes mellitus (NDM) and diabetes associated with extra-pancreatic features [2].

The genetic confirmation of diabetes in each clinical case is of crucial importance for personalized patient management: it determines drug therapy and allows the prediction of the clinical course and the timely diagnostic of associated diseases [3]. Among the monogenic diabetes associated with extra-pancreatic features, there are diseases with an autosomal dominant (HNF1B-MODY and GATA6 syndrome) or autosomal recessive type of inheritance (DIDMOAD, Walcott–Rallison syndrome, and Bardle–Biedl syndrome), insulin resistance type A, and diabetes inherited through the maternal line (mitochondrial diabetes) [2,4,5,6,7,8,9,10,11].

GATA6 syndrome is a rare disorder caused by pathogenic variants in the *GATA6* gene, which plays an important role in the differentiation of the mesoderm and visceral endoderm. The main components of *GATA6* syndrome are diabetes mellitus, exocrine pancreatic insufficiency, and developmental abnormalities of the digestive and cardiovascular systems [12,13,14]. Monogenic diabetes due to *GATA6* mutation often manifests as neonatal diabetes and abnormal pancreatic development; however, this syndrome may be diagnosed in older children or even adults, with or without pancreatic developmental defects, combined with heart defects, hepatobiliary defects, and abnormalities such as muscle hernia, growth retardation, hypothyroidism, etc. [15].

Worldwide, roughly 50 cases of monogenic diabetes associated with *GATA6* have been described [6], and only one clinical case has been previously described in the Russian Federation up to this point [16].

We report a novel heterozygous variant of a 4-nucleotide deletion (c.1302+4_1302+7del) in the donor splicing site of exon 3 of the *GATA6* gene in a patient of the age of 8 with neonatal diabetes, exocrine pancreatic insufficiency, gallbladder atresia, and a congenital heart defect. Our results of the long-term observation of a child with GATA6 syndrome with neonatal transient hyperglycemia that recurred in childhood expand the knowledge about the genetic and clinical features of patients with GATA6 syndrome. This will allow physicians to detect its pathology and conduct genetic analysis in a timely manner to facilitate a multidisciplinary approach for the treatment of patients with GATA6 syndrome.

## 2. Case Description 

### 2.1. Materials and Methods

Laboratory diagnostics included the study of glycated hemoglobin levels, hemoglobin, total and ionized calcium, phosphorus, fasting blood glucose, triglycerides, total cholesterol, low-density lipoprotein, high-density lipoprotein alanine aminotransferase and aspartate aminotransferase. The study of the hormonal profile included the determination of the insulin level (IRI), thyroid hormone (TSH), free thyroxine level (free T4), and insulin-like growth factor 1 (IGF1). Immunological diagnostics included autoantibodies to insulin (IAA), glutamic acid decarboxylase (GADA), islet cell cytoplasmic autoantibodies (ICA), tyrosine phosphatase-like protein IA-2 (IA-2A), zinc transporter 8 (ZnT8A), thyroid peroxidase antibodies (anti-TPO), and thyroglobulin antibodies (anti-Tg).

The study was approved by the Bioethics Committee of the National Medical Research Center of Endocrinology, protocol No. 16 dated 13 September 2023.

Genomic DNA was isolated from peripheral blood lymphocytes using a MagPure Blood DNA kit (Magen, Guangzhou, China). Quantitative and qualitative analyses of isolated DNA were performed using a Nanodrop 2000 spectrophotometer (Thermo Fisher Scientific, Waltham, MA, USA), a Qubit 2.0 fluorimeter (Invitrogen, Carlsbad, CA, USA), the Qubit dsDNA HS Assay Kit and a Nanodrop 2000 spectrophotometer (Thermo Fisher Scientific, Waltham, MA, USA).

Full-exome libraries were prepared using the KAPA HyperPlus Kit (Roche, Basel, Switzerland) following the manufacturer’s protocol to ensure high efficiency in library preparation and minimize DNA fragmentation errors. The enrichment of libraries for exonic regions was performed using the KAPA HyperExome Kit (Roche, Basel, Switzerland). The sequencing was conducted on an Illumina Novaseq 6000 platform (Illumina, San Diego, CA, USA), using the Novaseq 6000 S4 Reagent Kit v1.5 (200 cycles) (Illumina, San Diego, CA, USA) for paired-end sequencing with a read length of 2 × 100 bp. Additionally, specific primers were designed to target exon fragments of the *GATA6* gene, and Sanger sequencing was performed on an AB3500 instrument (Applied Biosystems, Thermo Fisher Scientific, Waltham, MA, USA) for further validation.

NGS data were processed using a bioinformatics pipeline optimized for high-throughput sequencing data. Reads were aligned to the human genome reference sequence (GRCh38) using the BWA-MEM algorithm, followed by post-alignment processing, including marking duplicates and base recalibration, to enhance variant calling accuracy. Variants were identified using the GATK (Genome Analysis Toolkit) (Broad Institute, Cambridge, MA, USA) and filtered based on established quality metrics (a minimum depth of 30× and quality score > 30). Variant annotation was performed for all known gene transcripts using the RefSeq database, and pathogenicity predictions were made following ACMG (American College of Medical Genetics and Genomics) guidelines.

For splicing variants, SpliceAI and AdaBoost algorithms were used to predict the functional impact of variants at splice sites and in adjacent intronic regions. To determine the clinical relevance of the identified variants, databases such as OMIM (Online Mendelian Inheritance in Man) and HGMD (Human Gene Mutation Database) were consulted. Additionally, software tools like Annovar (http://www.openbioinformatics.org/annovar/ (accessed on 1 October 2024)), SIFT (https://sift.bii.a-star.edu.sg/, accessed 1 October 2024), MutationTaster (online version (https://www.mutationtaster.org/, accessed 1 October 2024)), and MutPred (version 2.0 (http://mutpred.mutdb.org/, accessed 1 October 2024)) were employed to evaluate the effect of the variants on protein structure and function, providing a comprehensive understanding of potential pathogenic mechanisms. Population frequency databases, including 1000 Genomes, Exome Aggregation Consortium (ExAC), dbSNP, and HGMDB, were also cross-referenced to assess the rarity of the variants.

### 2.2. Results

A 3.5-year-old girl was admitted to our clinic with hyperglycemia and a short stature. Her mother’s pregnancy was complicated by chlamydia, acute respiratory infections, mild anemia, an operative delivery at 35 weeks (oligohydramnios), and intrauterine growth retardation: length 40 cm (SDS = −2.65) and weight 1580 g (SDS = −2.29). There was no family history of endocrine diseases and the parents were non-consanguineous.

Hyperglycemia up to 29.6 mmol/L was detected within the first week of her life, and insulin therapy (regular insulin and detemir) was initiated. Insulin therapy was discontinued a month later, and normoglycemia was maintained. The girl was diagnosed with transient NDM, and anomalies in the development of the digestive tract (agenesis of the gallbladder), a minor anomaly of the heart (mitral valve prolapse), and an umbilical hernia were also found. At the age of 2 years, glycosuria was detected. At the age of 2 years, clinical symptoms of diabetes appeared (polyuria, polydipsia, and a lack of weight gain), as well as glycated hemoglobin (HbA1c)—11.5%—and postprandial glycemia 18.6 mmol/L. Insulin therapy (in part) was initiated.

At the age of 3.5 years, the following conditions were discerned: a height of 86.9 cm (SDS: −1.95), a body weight of 10.5 kg (SDS BMI: −1.85), an umbilical hernia, and abdominal distension. Insulin administered as a partial dose was 0.45–0.6 U/kg/day. Glycemic control was good—HbA1c 5.8%—the fasting C-peptide level was slightly below the reference range (0.69 ng/mL), and maximum C-peptide level during a mixed-meal test—2.32 ng/mL (Table 1). 

Her complete blood count, IGF-1 level and biochemical parameters were within normal ranges. The results of the laboratory examinations are presented in Table 2.

Islet antibodies (GADA, ICA, IAA, IA-2A, and ZnT8A) were negative. According to the ultrasound examination, the gallbladder was not visualized, the pancreas was visualized fragmentarily (due to flatulence), and a hernia sac (1.2 × 0.6 cm) without contents was located in the navel area. No structural pathology of the kidneys was detected.

Genetic testing with next-generation sequencing (NGS) of the 27 genes most commonly associated with monogenic diabetes genes (*GCG*, *GLUD1*, *WFS1*, *HNF1A*, *GCK*, *INS*, *HNF1B*, *ABCC8*, *HNF4A*, *RFX6*, *PTF1A*, *AKT2*, *ZFP57*, *INSR*, *EIF2AK3*, *PPARG*, *PAX4*, *PDX1*, *GLIS3*, *KCNJ11*, *SLC16A1*, *FOXP3*, *BLK*, *CEL*, *KLF11*, *SCHAD*, and *GCGR*) was performed but without significant results.

At the age of 8, the patient had dyspeptic symptoms (bloating and heaviness), a loose stool and undigested food in her stool. The girl was diagnosed with autoimmune thyroiditis while she was experiencing euthyroid (Table 2).

Taking into account the clinical presentation and previous negative results of the NGS panel, whole-exome sequencing was performed. In the *GATA6* gene (NC 000018.10 (nm 005257.6) in the donor site of the splicing of exon 3, a variant (hg38, chr18:22177125_22177128del, c.1302+4_1302+7del), previously undescribed in the literature, was found in the heterozygous state, leading to the deletion of four nucleotides with a coverage depth of 125×. The variant was not found in the population frequency database GNOMAD and has not been described in the literature. The variant is located in a conservative position after exon 3 of the gene and presumably leads to the activation of additional splicing sites. The computer algorithm for predicting the effect of nucleotide variants on the function splice sites (SpliceAI/varSEAK) characterizes this variant as pathogenic. Pathogenic splicing variants in this region of the gene have been previously reported, such as c.1303-10C>G [17] and c.1303-1G>T [18]. The discovered variant may affect the functioning of two tandem zinc fingers of GATA protein and disrupt the DNA binding domain.

This variant was not found in the child’s parents. According to the literature, cases of de novo variants in the *GATA6* gene are a frequent event, reaching 73.6% of all cases [6].

Exocrine pancreatic insufficiency is typical for patients with pathogenic variants of the *GATA6* gene, and the level of fecal elastase-1 was studied, and a decrease in the indicator was found (35 μg/g, with a normal level of >200).

Pancreatic enzyme replacement therapy was started, resulting in a reduction in dyspeptic symptoms and increased growth rate (Figure 1)—during a dynamic examination at 9 years, moderate growth abnormalities persisted (growth SDS −1.82) with normal growth rates (growth rate 5.73 cm/year and growth rate SDS +0.38) (Table 3). The predicted growth corresponds to the target height of 160.5 cm (Figure 1).

A rapid progression of myopia was diagnosed (at the age of 8 years, Sph −1.5 OD and OS, and at the age of 9 years, Sph −5.5 OD and Sph −4.75 OS).

## 3. Discussion

We presented the clinical case of a patient diagnosed with NDM that remitted within months and who relapsed at 2 years. The most common causes of transient NDM are pathogenic variants in the genes encoding proteins of ATP-dependent potassium channels, the insulin gene, and the pathology of chromosome 6. Rarely, the condition is caused by functional defects of β-cells (*GCK*), the destruction of β-cells due to apoptosis (*EIF2AK3*), autoimmune processes (FOXP3-IPEX syndrome), and defects of transcription factors (*IPF1*, *GLIS-3*, *SLC19A*, *SLC2A2*, *NFT1B*, *PTFA1*, *GATA6*, *GATA4*, *and NEUROD1*) [19,20]. There have been 50 clinical cases of GATA6 syndrome reported recently, 60% of them with pancreatic developmental anomalies [21]. GATA6 syndrome is characterized by a combination of neonatal diabetes mellitus (up to 72.7% of cases [6]), exocrine pancreatic insufficiency (up to 67.3% of cases), and congenital developmental anomalies including the heart (up to 89.9% of cases) and the gastrointestinal tract [6,12,13,14]. The *GATA6* gene is mapped to 18q11.2, consists of seven exons and is a member of a family of zinc finger transcription factors [13]. The GATA6 protein plays an important role in the early stages of the differentiation of pancreatic acini and is also involved in the development of organs of endodermal and mesodermal origin, including the intestine, lungs, gonads, heart and pancreas [22,23], and it interacts with tissue-specific transcription factors and promotes the differentiation of cell types [23].

Initially, this gene was described in the context of cardiac embryogenesis [24] and patients with congenital heart defects [17,25]. Its association with agenesis, hypoplasia of the pancreas and diabetes mellitus was reported later [25]. Currently, heterozygous inactivating variants in *GATA6* are the most common cause of pancreatic developmental anomalies [26].

Due to limited descriptions of different variants, it is hard to assess genotype–phenotype associations [27]. The prevalence of pathogenic variants, as well as the functional role of GATA family proteins, is unknown; the low incidence of defects in the *GATA6* gene is probably associated with high embryonic mortality, which has been experimentally proven [21,28].

A phenotype of the syndrome can be heterogeneous in relatives with a more severe clinical course in the children of affected parents [29,30]. De novo variants in this gene occur in 73.6% of all syndrome cases [6], and in most cases of GATA6 syndrome, the proband does not have a family history of diabetes mellitus. There is evidence of a development of NDM as a result of pancreatic agenesis in patients with de novo variants [6], but in our case, there was no pancreatic agenesis.

### 3.1. Diabetes Mellitus Associated with GATA6

The development of diabetes mellitus with heterozygous variants in the *GATA6* gene is caused by premature apoptosis, impaired islet cell proliferation, an abnormal structure of the endoplasmic reticulum and functional failure of β-cell mitochondria, an accumulation of immature insulin, and a decreased production and secretion of insulin [31,32]. Due to the decreased expression of transcription factors that promote endocrine function and pancreas development, *PDX1*, *MAFA*, and *NKX6.1* [15], pancreatic agenesis is present in 83.6% [6]. The age of dysglycemia/diabetes manifestation varies, even within the same family [29].

According to some authors, pathological *GATA6* gene variants may account for 3% of all NDM cases. In GATA6 syndrome, NDM may have a transient and permanent clinical course. In the study of E. De Franco et al., hyperglycemia or diabetes mellitus in adulthood in combination with heterozygous variants in the *GATA6* gene presented in the parents of the examined patients [18]. Relapses of diabetes are possible in childhood or adulthood [25,26,33]. Our case is only the fourth description of the recurrence of transient NDM in childhood [6,29,33,34].

Diabetes mellitus with pathological variants in the *GATA6* gene, as in our case, usually requires low doses of insulin (0.2–0.35 U/kg/day), which correlates with the β-cell mass [18], and treatment with sulfonylurea in GATA6 syndrome is unsuccessful [15].

### 3.2. Other Components of GATA6 Syndrome

In this case, the patient had moderate stunting compared to their peers from birth. Due to insulin deficiency with pathogenic variants in the GATA6 gene, as in other monogenic causes of NDM [14], the level of insulin-like growth factor 1 (IGF-1) decreases in the prenatal period, causing IUGR [13,27,29]. The subsequent growth of patients is normal [26] or below average [29], which also may be associated with nutritional deficiency due to exocrine pancreatic insufficiency, a common component of GATA6 syndrome [35]. There may be signs of the insufficient absorption of micro- and macronutrients, such as being underweight and having a short stature [36], mineral metabolism disorders, vitamin deficiency [37], etc., which can also be the cause of postnatal short stature.

The identification of the pathogenic variant in *GATA6* allowed us to perform a diagnosis of exocrine pancreatic insufficiency. According to the literature, the early diagnosis and therapy of exocrine pancreatic insufficiency and insulin therapy had a positive effect on the growth prognosis in patients with *GATA6* gene variants [38] and was confirmed in our clinical case.

Our patient was diagnosed with agenesis of the gallbladder, which is associated with GATA6 syndrome. Other malformations of the digestive system have been described in patients with GATA6: biliary atresia, intestinal malrotation, and protein-losing enteropathy [25]. In 2023, necrotizing enterocolitis with intestinal perforation was described for the first time [27].

Congenital heart defects (atrial and ventricular septal defects [15,18,29], the tetralogy of Fallot [39], tricuspid valve atresia, pulmonary artery stenosis, and the transposition of the great arteries [13,25]) are described in up to 93% of patients with GATA6 syndrome [17]. In the described clinical case, the patient was diagnosed with mitral valve prolapse.

Among other components of this syndrome, such as a diaphragmatic hernia, which was observed in our case [29], neurocognitive disorders [25,38], hypogonadism, and congenital hypothyroidism [39] are described. Our patient was diagnosed with chronic autoimmune thyroiditis and progressive myopia. There are no descriptions of visual impairment associated with variants in the *GATA6* gene in the literature, but it is known that *GATA6* as a transcription factor is involved in the development of eye structures [40].

During the genetic counseling of the patient’s family, the parents reported no family history of diabetes, and genetic testing revealed no changes in the *GATA6* gene in the parents, whereby a described pathogenic variant in the child arose de novo. Thus, the recurrence risk of the disease is 50% for the offspring and almost negligible for siblings of the proband.

### 3.3. Future Directions

Future research should be focused on the functional characterization of the identified GATA6 variants to understand their impact on pancreatic development and endocrine function. While the direct editing of the identified deletion using CRISPR/Cas9 may be challenging, the technology can be leveraged to create cell and animal models harboring the same deletion. These models are instrumental in investigating the specific molecular pathways affected by the *GATA6* loss of function, including its impact on beta cell differentiation and insulin secretion.

Further, CRISPR/Cas9-mediated gene editing can be used to explore potential corrective strategies by introducing repair templates that restore the wild-type sequence through homology-directed repair (HDR). Alternatively, patient-derived induced pluripotent stem cells (iPSCs) with the *GATA6* deletion can be used to develop pancreatic organoids, providing a platform for studying the functional consequences of the mutation in a controlled environment.

By conducting these studies, researchers can elucidate the mechanisms by which *GATA6* mutations disrupt pancreatic function and potentially identify therapeutic targets for developing novel interventions for monogenic diabetes associated with *GATA6* variants.

## 4. Conclusions

GATA6 syndrome is a very rare disease with an autosomal dominant inheritance caused by pathogenic heterozygous variants in the GATA6 gene, which is characterized by a combination of diabetes mellitus (predominantly in the neonatal period), exocrine pancreatic insufficiency, and congenital heart and gastrointestinal defects. The presented clinical case demonstrates the feasibility of whole-exome sequencing in patients with non-immune diabetes in combination with congenital extra-pancreatic features (the agenesis of the gallbladder, congenital heart disease, an umbilical hernia, etc.). In our case, the cause of NDM was not identified initially with sequencing most common diabetes genes, while whole-exome sequencing revealed a pathogenic, heterozygous variant in the GATA6 gene.

In conclusion, it should be emphasized that all children with NDM are recommended to be referred to a genetic verification of their diagnosis. In case of multiple congenital defects in combination with diabetes mellitus, genetic testing should take into account rare genetic variants, and one effective approach is whole-exome sequencing. Patients with GATA6 syndrome need dynamic follow-up carried out by a multidisciplinary team (pediatricians, pediatric endocrinologists, gastroenterologists, and cardiologists) for the early detection of components of the syndrome.

A referral to a specialist in monogenic diabetes or an interested clinical genetics unit is suggested to guide specific management considerations and/or facilitate genetic testing of other related affected or pre-symptomatic individuals.

## Figures and Tables

**Figure 1 ijms-25-11998-f001:**
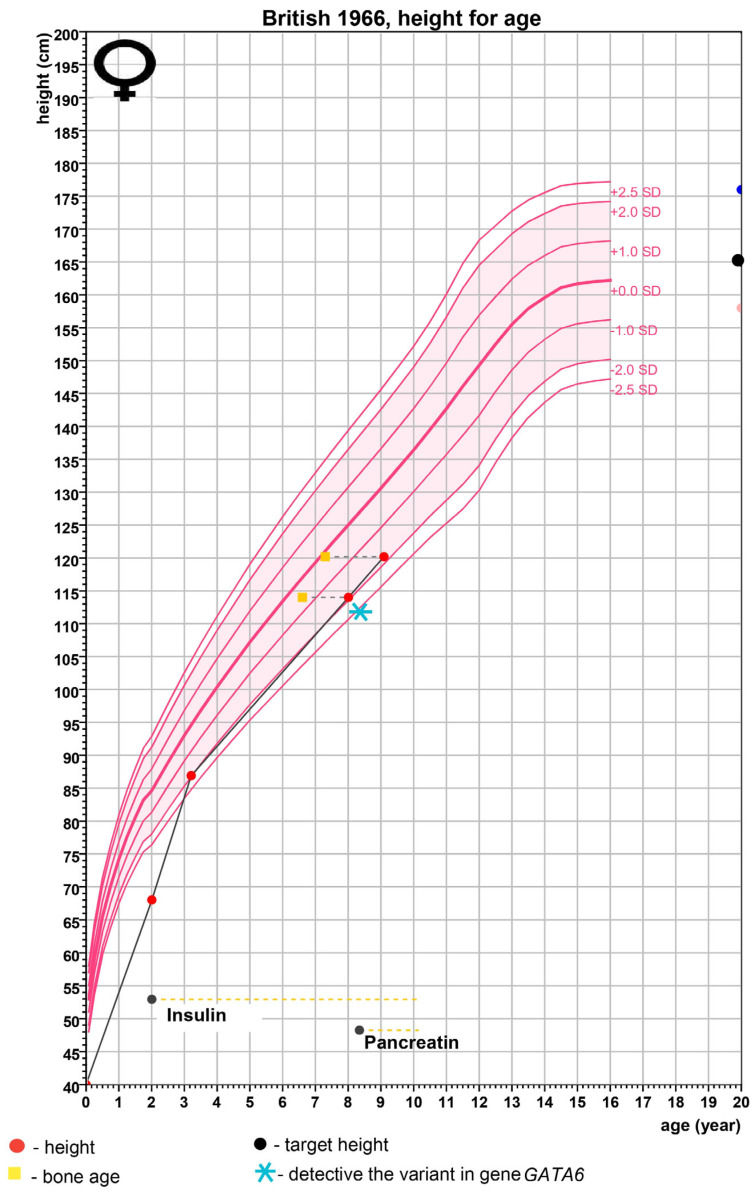
Patient’s growth chart. Growth chart for the children with GATA6 syndrome.

**Table 1 ijms-25-11998-t001:** Results of mixed-meal test at age 3.5 and 8 years, with duration of the disease 6 months and 5.5 years, respectively.

Time, min	3.5 Years (Duration of the Disease 6 Months)		8 Years (Duration of the Disease 5.5 Years)	
	Blood Glucose, mmol/L	C-Peptide, ng/mL	Blood Glucose, mmol/L	C-Peptide, ng/mL
0	4.99	0.69	5.69	0.71
30	-	-	10.96	1.96
60	-	-	8.7	1.97
90	-	-	6.78	1.37
120	6.34	2.32	5.81	0.931

**Table 2 ijms-25-11998-t002:** Clinical and laboratory data of patient M.

Parameter	3 years	8 years	9 years
Height, sm	86.9	114	120.2
SDS height	−1.95	−1.91	−1.82
Weight, kg	10.5	18	20
SDS BMI	−1.92	−1.47	−1.62
Insulin dose, U/kg/day	0.45–0.6	0.9	0.9
Carbohydrate Metabolism Parameters			
HbA1c, %	5.8	5.2	5.6
fasting blood glucose, mmol/L	4.99	5.69	6.64>
fasting C-peptide, ng/mL (N: 1.1–4.4)	0.69<	0.71<	0.849
fasting insulin, µU/mL (N: 2.3–26.4)	3.39	2.2	3.36
Other Laboratory Parameters			
TSH, mIU/L, (N: 0.64–5.76)	1.61	1.75	5.165>
Free thyroxine, pmol/L (N: 11.5–20.4)	16.24	12.64	11.5
IGF-1, ng/mL (N: 8–290)	67.93	81.47	74.3
Anti-Tg, IU/mL (N: 0–64)	-	18,540>	5634>
Anti-TPO, IU/mL (N: 0–5.6)	-	1.78	91.46>
Cortisol, nmol/L (N: 77–630)	-	453.3	508.6

**Table 3 ijms-25-11998-t003:** Anthropometric parameters during examinations.

Age, Years	Event	Height, cm	SDS Height	Growth Rate, cm/year	SDS Growth Rate	Weight, kg	SDS BMI	Bone Age, Years
3.5	Recurrence of NMD Initiation of insulin therapy	86.9	−1.95	-	-	10.5	−1.59	-
8.01	Detection of the variant in gene GATA6 Initiation replacement therapy with pancreatic enzymes	114.0	−1.92	-	-	18.0	−1.40	6.6
9.09		120.2	−1.82	5.7	+0.33	20.0	−1.57	7.3

## Data Availability

The data from this study can be obtained from the corresponding author upon making a reasonable request if there are no privacy or ethical issues.

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
