# Peer review of "The Identification of a Novel Pathogenic Variant in the GATA6 Gene in a Child with Neonatal Diabetes"

_ijms, 2024, doi:10.3390/ijms252211998_

Round 1
Reviewer 1 Report
Comments and Suggestions for Authors
An interesting case report in which the authors present a novel variant in the GATA 6 gene in a child with neonatal diabetes.
The manuscript is well-structured, adhering to the journal's standards. However, I have a few minor suggestions that could enhance the quality of the manuscript.
Keywords are important tools for enhancing the visibility of the article in the event of its publication. Some of the keywords used for this manuscript are too vague, so I suggest adding a few additional ones that are more specific to the topic.
Exclusion criteria should also be included in the Material and Methods.
At the end of the discussion section, I suggest the authors to include the limitations of the study and the strenghts of the study. Additionally, I suggest adding a paragraph before the final conclusions, outlining future research directions to guide other researchers in the field.
A table with all the abbreviations and their eplanations would be helpful.
Some of the references are older than 10 years. I suggest replacing them with novel research in the field.
Author Response
Thank you for your reply and kind! We have taken the edits into account and have significantly revised our manuscript in all sections.
Reviewer 2 Report
Comments and Suggestions for Authors
I would like to thank Sechko et al for this interesting case report describing a novel pathogenic variant in the GATA6 gene 2 in a child with neonatal diabetes. Unfortunately, despite being an interesting medical case, the overall quality of the paper is rather low. Initially, an extensive improvement of English is crucial. In many sentences, no verb is present, especially when it comes to passive voice, rendering the comprehension difficult. In addition, basic medical mansucript principles are lacking: genes should be written in italics, abbreviations should correspond to the word used (for example why use T4sv as an abbreviation for free thyroxine level?) and be described when first mentioned in the manuscript, discussion should relate paper's findings with the previous literature and so forth. Further, some more specific comments:
In the abstract, the idea is to start from more general information and to continue with presenting the current case. You start from the first sentence with describing your case which is wrong.
In Materials and Methods, you describe something totally different to a case report: about a sample of 507 children that were examined and genetically evaluated. Which one is true? In addition, you mention twice that "Written informed consent was obtained from each patient/patient representative." If it is a retrospective study and you examined medical files, how did you manage to obtain written informed consent?
Line 154: what is lientorrhea?
The presentation of the clinical progression of the child is rather confusing. The reader cannot follow exactly what happened as the child grew. Please try to be more concise and to the point.
In Discussion and Conclusion, please try to correlate your findings with existing literature without repeating the same data.
Comments on the Quality of English Language
Must be corrected throughout the manuscript by a native English speaker. Makes the manuscript difficult to comprehend.
Author Response
Thank you for your kind! We have taken the edits into account and have significantly revised our manuscript in all sections.
Reviewer 3 Report
Comments and Suggestions for Authors
This article presents a clinical case of neonatal diabetes associated with a novel pathogenic variant in the GATA6 gene. The case provides insights into the genetic etiology and phenotypic manifestations of GATA6 syndrome, which is a rare and complex disorder involving diabetes mellitus, exocrine pancreatic insufficiency, and various congenital abnormalities. While the paper offers valuable clinical and genetic data, there are several areas that require improvement or clarification to meet the standards of high-impact journals.
Strengths:
- Unique Case Presentation: The article describes a previously unreported variant in the GATA6 gene, expanding the existing knowledge of the genetic mutations associated with GATA6 syndrome.
- Comprehensive Clinical Evaluation: The paper provides a detailed clinical history, laboratory findings, and genetic analysis, offering a complete picture of the patient's condition.
- Genetic Methodology: The use of next-generation sequencing (NGS) and Sanger sequencing to confirm the variant adds scientific rigor to the case.
Key Areas for Improvement:
- Introduction Clarity and Focus:
- The introduction could benefit from clearer delineation of the topic. While it effectively sets the context of monogenic diabetes, it should more explicitly emphasize the novel contribution of the paper—specifically, the identification of the new GATA6 variant.
- Consider elaborating on the significance of GATA6-related syndromes in a broader clinical context (e.g., the frequency and impact of missed diagnoses in such rare cases).
- Literature Review:
- The literature review does not thoroughly address the existing body of work on GATA6 variants, especially regarding genotype-phenotype correlations. A more detailed comparison with previously reported cases would strengthen the novelty of the new variant presented here.
- The section discussing GATA6 syndrome lacks sufficient depth in reviewing alternative pathogenic variants and the spectrum of clinical manifestations. Adding more references to prior studies could help ground the findings within the existing literature.
- Case Description (Materials and Methods):
- The article presents itself as a case report focused on a single patient with a novel pathogenic variant in the GATA6 gene, yet the Materials and Methods section mentions a sample of 507 patients with suspected hereditary forms of diabetes. This creates some confusion regarding the scope of the study. If the primary focus is a case report, the reference to the larger cohort should be clarified. It would be helpful to explain how the case described fits within the broader context of the 507 patients and whether similar genetic or clinical features were observed in other patients in the cohort. Additionally, providing insight into how this specific case was selected for detailed reporting would strengthen the clarity and focus of the article.
- The clinical case is well-presented, but it could be more concise. Certain elements, such as the repeated mention of minor congenital anomalies (umbilical hernia), could be shortened to focus on more relevant clinical findings (e.g., pancreatic insufficiency and diabetes management).
- The methods used for genetic analysis should be described in more detail, particularly regarding the rationale for selecting NGS and the specific bioinformatics tools employed. A flowchart summarizing the genetic workflow would enhance clarity.
- Discussion:
- The discussion should more effectively relate the findings of the case to broader implications for clinical practice. How does the identification of this new variant influence future diagnosis or treatment strategies for GATA6 syndrome?
- The discussion on the patient’s growth and development feels somewhat disconnected from the genetic findings. Consider elaborating on the link between the identified mutation and its effect on insulin secretion and growth regulation.
- The conclusion could be strengthened by suggesting potential next steps, such as follow-up studies or the utility of early genetic screening in similar cases.
- Tables and Figures:
- The clinical data tables are informative but could be simplified. For instance, a visual representation of the patient's growth trajectory (beyond Figure 1) could be more impactful if accompanied by an interpretation of the growth data in relation to the genetic findings.
- Ethical Considerations:
- While the ethical approval and informed consent processes are mentioned, the article should briefly address how privacy and confidentiality were maintained, especially given the potential for re-identification in rare genetic cases. Furthermore, should be addressed this statement “Informed consent was obtained from all subjects involved in the 309 study. Written informed consent has been obtained from the patient's parents to publish this paper.”
- Language and Grammar:
- There are minor grammatical issues throughout the text. For example, sentences like "The patient also had concomitant diseases that cannot be clearly linked to the pathogenic variant in the GATA6 gene" could be more concisely phrased.
- Improving the overall language quality would enhance the readability of the paper. Consider having the manuscript reviewed by a professional language editing service.
Additional Recommendations:
- Clinical Relevance: Include a paragraph discussing how this case contributes to improving diagnostic tools or guidelines for pediatricians and endocrinologists handling monogenic diabetes.
- Future Directions: The paper would benefit from a section on future research, especially regarding the functional analysis of GATA6 variants and potential targeted therapies.
- Percent Match: 24% from iThenticate report should be addressed to be under 20%.
In summary, the article presents an interesting and rare case of GATA6 syndrome, but it requires further refinement to meet the high standards of a prestigious journal. Specifically, the introduction, discussion, and integration of the literature review should be strengthened, and minor grammatical issues should be addressed. With these revisions, the paper could make a valuable contribution to the field of pediatric endocrinology and genetics.
Comments on the Quality of English Language- There are minor grammatical issues throughout the text. For example, sentences like "The patient also had concomitant diseases that cannot be clearly linked to the pathogenic variant in the GATA6 gene" could be more concisely phrased.
- Improving the overall language quality would enhance the readability of the paper. Consider having the manuscript reviewed by a professional language editing service.
Round 2
Reviewer 2 Report
Comments and Suggestions for Authors
I can't find a point by point reply to my uploaded comments and suggestions.
Comments on the Quality of English LanguageNo comments
Author Response
I am very sorry that you have not found a response to your uploaded comments. Please see the attachment.

Reviewer 3 Report
Comments and Suggestions for Authors
Additional recommendations were not addressed in the previous revisions:
- Clinical Relevance: Include a paragraph discussing how this case contributes to improving diagnostic tools or guidelines for pediatricians and endocrinologists handling monogenic diabetes.
- Future Directions: The paper would benefit from a section on future research, especially regarding the functional analysis of GATA6 variants and potential targeted therapies.
- Percent Match: 24% from iThenticate report should be addressed to be under 20%.
Author Response
Comments 1: Clinical Relevance: Include a paragraph discussing how this case contributes to improving diagnostic tools or guidelines for pediatricians and endocrinologists handling monogenic diabetes.
Response 1: We agree with this comment. We have edited this section. This change can be found – page number 8, line number 280-285.
Comments 2: Future Directions: The paper would benefit from a section on future research, especially regarding the functional analysis of GATA6 variants and potential targeted therapies.
Response 2: We agree with this comment. We added a new section about future directions. This change can be found – page number 7, line number 252-269.
Round 3
Reviewer 2 Report
Comments and Suggestions for Authors
Thank you.